# Silica Coated Bi_2_Se_3_ Topological Insulator Nanoparticles: An Alternative Route to Retain Their Optical Properties and Make Them Biocompatible

**DOI:** 10.3390/nano13050809

**Published:** 2023-02-22

**Authors:** Blaž Belec, Nina Kostevšek, Giulia Della Pelle, Sebastjan Nemec, Slavko Kralj, Martina Bergant Marušič, Sandra Gardonio, Mattia Fanetti, Matjaž Valant

**Affiliations:** 1Materials Research Laboratory, University of Nova Gorica, 5000 Nova Gorica, Slovenia; 2Department for Nanostructured Materials, Jožef Stefan Institute, 1000 Ljubljana, Slovenia; 3Jožef Stefan International Postgraduate School, 1000 Ljubljana, Slovenia; 4Department for Material Synthesis, Jožef Stefan Institute, 1000 Ljubljana, Slovenia; 5Faculty of Pharmacy, University of Ljubljana, 1000 Ljubljana, Slovenia; 6Laboratory for Environmental and Life Sciences, University of Nova Gorica, 5000 Nova Gorica, Slovenia

**Keywords:** topological insulator, bismuth selenide, photo-thermal material, biocompatibility, nanoparticles

## Abstract

Localized surface plasmon resonance (LSPR) is the cause of the photo-thermal effect observed in topological insulator (TI) bismuth selenide (Bi_2_Se_3_) nanoparticles. These plasmonic properties, which are thought to be caused by its particular topological surface state (TSS), make the material interesting for application in the field of medical diagnosis and therapy. However, to be applied, the nanoparticles have to be coated with a protective surface layer, which prevents agglomeration and dissolution in the physiological medium. In this work, we investigated the possibility of using silica as a biocompatible coating for Bi_2_Se_3_ nanoparticles, instead of the commonly used ethylene-glycol, which, as is presented in this work, is not biocompatible and alters/masks the optical properties of TI. We successfully prepared Bi_2_Se_3_ nanoparticles coated with different silica layer thicknesses. Such nanoparticles, except those with a thick, ≈200 nm silica layer, retained their optical properties. Compared to ethylene-glycol coated nanoparticles, these silica coated nanoparticles displayed an improved photo-thermal conversion, which increased with the increasing thickness of the silica layer. To reach the desired temperatures, a 10–100 times lower concentration of photo-thermal nanoparticles was needed. In vitro experiments on erythrocytes and HeLa cells showed that, unlike ethylene glycol coated nanoparticles, silica coated nanoparticles are biocompatible.

## 1. Introduction

In the last two decades, topological insulator nanoparticles (TI) have been intensively studied due to their attractive electronic properties, resulting from their metallic, linearly dispersing, and spin-polarized topological surface states (TSS). A TI is like an ordinary insulator in its interior, but it contains protective conductive TSS at the material’s borders (surface or edges) [1,2,3,4]. The presence of the surface conduction electrons in TSS, which can resonate upon optical excitation, is considered to be the origin of the localized surface plasmon resonance (LSPR) displayed by TI in the low-frequency part of the visible range (>400 nm) [5,6,7]. Moreover, due to this LSPR, nanoparticles display a photo-thermal effect (PE), which makes them relevant in the field of medical diagnosis and therapy [6,8,9,10,11,12,13,14].

A typical representative and one of the most promising TIs, due to a narrow bulk band gap of 0.3 eV, is bismuth selenide (Bi_2_Se_3_) [15,16,17]. To fully exploit the exotic properties of Bi_2_Se_3_ nanoparticles, it is crucial to prepare them without an adsorbent on their surface, which could mask or quench their surface and the optical properties originating from TSS [7]. 

Despite the interesting properties that make TI nanoparticles relevant for numerous applications, adsorbent-free TI nanoparticles are not suitable for biomedical applications. Nanoparticles without any protective surface layer will agglomerate immediately when they are subjected to a physiological medium (e.g., serum, cell culture, blood). In addition, the protective surface layer also counteracts the possible dissolution of the nanoparticles and prevents the leakage of potentially toxic metal ions. The surface layer can be used for further functionalization [18,19,20,21,22]. 

There is a scarcity of reports regarding the application of Bi_2_Se_3_ nanoparticles in biomedicine and their effects in vivo and in vitro [6,8,9,10,11,12,13,14]. A major challenge is to prevent their oxidation and instability, which limits their practical applications [13]. To date, this problem has been solved by preparing Bi_2_Se_3_ nanoplatelets through the solvothermal method, using ethylene-glycol (EG) and polyvinylpyrrolidone (PVP) as the solvent and capping agent, respectively. Reports claimed that stability and biocompatibility were achieved using PVP molecules attached to the surface of Bi_2_Se_3_ nanoparticles [9,12,13,23,24,25,26,27,28]. However, in 2018, we showed that the solvothermal method, where EG and PVP were used, resulted in Bi_2_Se_3_ nanoplatelets only coated with an approximately 2 nm thick EG layer. This was confirmed by ζ-potential measurements and TG-MS analyses. These analyses did not show the presence of PVP [7]. 

Since it was shown that Bi_2_Se_3_ nanoparticles produced via the solvothermal method are coated with EG, which is toxic, this raises safety concerns for using such Bi_2_Se_3_ nanoparticles in biomedical applications. Despite the beneficial properties of EG, such as reducing the surface charges and therefore improving the colloidal stability in the physiological environment, prolonging the nanoparticles’ half-life and preventing potential dissolution, EG can affect the central nervous, cardiopulmonary, and renal systems, and its metabolites can cause numerous deleterious effects at the cellular level, e.g., oxidative stress [29,30,31,32]. In addition, the Food and Drug Administration (FDA) only approved EG for use indirectly, as a component of packaging adhesives in the food industry [33]. Moreover, further functionalization of EG-coated Bi_2_Se_3_ nanoparticles is challenging, because it often requires chemicals that are expensive, toxic, or unstable, and the synthetic procedures require personnel with the appropriate laboratory skills (e.g., human serum albumin, doxorubicin—a drug for treating cancer, etc.) [10,11]. 

Based on the described drawbacks of using EG as a coating, there is a need to prepare Bi_2_Se_3_ nanoplatelets coated with a non-toxic, stable, and biocompatible layer that allows further easy functionalization and does not mask the optical properties of Bi_2_Se_3_, as EG does. The most similar molecule, and less toxic than EG, is polyethylene glycol (PEG), which can be used as a coating and is already the gold standard for biocompatible coatings. PEGylation of nanoparticles is commonly used in liposomes and other nanoparticles to increase the circulation time and slow the rate of elimination by the reticuloendothelial system. However, despite the widespread use of PEG in food and drugs, people have started to show an immune response by developing anti-PEG antibodies, which can lead to the rapid elimination of PEG-coated nanoparticles [34,35,36,37]. 

Therefore, one solution to replace EG or PEG could be porous amorphous silicon dioxide (silica). Unlike EG, silica is approved by the Food and Drug Administration (FDA) and European Food Safety Authority (EFSA) as non-toxic and biodegradable, with an uptake of up to 1500 mg/day [38,39,40]. Moreover, it has an excellent porous structure, adjustable pore size, and easily modified surface [38,41,42,43,44]. Due to these properties, silica nanoparticles have great potential for biomedical applications (i.e., cancer therapy, DNA transfection, drug delivery, dental medicine, regenerative medicine, etc.) [45,46,47,48]. As a coating material, silica is used in combination with various functional nanomaterials [49,50], i.e., magnetic nanoparticles [51,52,53,54,55,56,57,58,59], luminescent nanoparticles [48,60,61], and photo-thermal responsive nanomaterials [49,56,62] etc., making them biocompatible and extending their applicability through further functionalization.

In this work, we successfully coated Bi_2_Se_3_ nanoparticles with layers of silica of different thicknesses. As mentioned earlier, a coating can mask or impair the optical properties of the TI nanoparticles derived from the TSS. Silica-coated Bi_2_Se_3_ nanoparticles retain these optical properties, except for those with the thickest, ≈200 nm, silica layer. Silica-coated nanoparticles exhibit good photo-thermal properties and specific adsorption rate (SAR) values. In vitro experiments showed that pristine (hydrothermally prepared) and silica-coated Bi_2_Se_3_ nanoparticles do not induce hemolysis and are not cytotoxic, even at very high concentrations of up to 1 g/L.

## 2. Materials and Methods

### 2.1. Materials

All the chemicals were used without further purification. 

Bi_2_Se_3_ nanoparticles synthesis: Bismuth (III) nitrate pentahydrate (Bi(NO_3_)_3_·5H_2_O), bismuth (III) oxide (Bi_2_O_3_, 99.98%), selenium powder (Se, ≥99.5%), hydrochloric acid (HCl), hydrazine hydrate (N_2_H_4_·H_2_O, 35%), sodium hydroxide (NaOH), ethylene glycol (C_2_H_6_O_2_, 99%), polyvinylpyrrolidone (PVP, MW = 8000). All chemicals were purchased from Alfa Aesar, Haverhill, MA, USA. 

Amorphous silica coating: Hexadecyltrimethylammonium bromide (CTAB) was obtained from Alfa Aesar (Kandel, Germany), absolute ethanol and aqueous ammonia (~25%) from Merck KGaA (Darmstadt, Germany), cyclohexane from VWR Int. GmbH (Vienna, Austria), tetraethoxysilane (TEOS), concentrated hydrobromic acid (~48%; HBr) and 2-amino-2-(hydroxymethyl)-1,3-propanediol (TRIS) from Sigma-Aldrich (St. Louis, MO, USA); 0.1 M HBr was prepared by diluting concentrated HBr (~48%) with deionized water. 

Hemolysis: Erythrocytes were isolated from whole sheep blood (BioSap SO, defibrinated, purchased from AdvaMed d.o.o., Ljubljana, Slovenia). Phosphate-Buffered Saline tablets (PBS) (Sigma Aldrich, St. Luis, MO, USA), potassium chloride (KCl), and sodium chloride (NaCl).

In vitro cell viability assay: Dulbecco’s modified Eagle’s medium (DMEM) (Gibco, Paisley, UK), fetal bovine serum (FBS) (Gibco, Paisley, UK), Penicillin Streptomycin (Gibco, Grand Island, NY, USA), 2.5% Trypsin (10X) (Gibco, Paisley, UK), phosphate-buffered saline (PBS) (Gibco, Paisley, UK), PrestoBlue Viability Reagent (Invitrogen, Eugene, OR, USA).

### 2.2. Methods

#### 2.2.1. Synthesis of Bi_2_Se_3_ Nanoparticles

The Bi_2_Se_3_ nanoparticles were prepared using the hydrothermal method, according to the procedure described in Ref. [7]. In short, stoichiometric amounts of bismuth nitrate (1 mmol) and selenium (1.5 mmol) were dissolved in 20 mL of deionized water under vigorous stirring, followed by the addition of 11 M HCl (25 µL) and hydrazine (1.6 mL). The obtained grey slurry was sealed in a Teflon-lined autoclave and heated at 240 °C for 48 h. Then the product was washed several times with deionized water. This sample was denoted as **“BiSe”**. Afterwards, the clean, adsorbent-free platelets were coated with amorphous SiO_2_ (silica) of different thicknesses. The coating procedures are described below. 

To compare and evaluate the effect of silica coating, the Bi_2_Se_3_ nanoparticles were also prepared by the solvothermal method, according to the procedure described in Ref. [63]. In brief, PVP and stoichiometric amounts of Bi and Se were dissolved in 20 mL of ethylene glycol. The grey slurry was sealed in a Teflon-lined autoclave and heated to 200 °C for 10 h. After the reaction had finished, the product was washed several times with deionized water. The solvothermally-synthesized platelets were denoted as **“BiSe ST”**.

#### 2.2.2. Coating of Bi_2_Se_3_ Nanoparticles with Amorphous Silica

*Thin silica coating:* First, a thin silica coating (2–5 nm) was deposited on the platelets. The procedure was based on Ref. [64]. In a glass beaker, 320 mg of CTAB was dissolved in 40 mL of deionized H_2_O. Then, the pH of the formed CTAB solution was adjusted to 3.30 using 0.1 M HBr. Separately, 200 mg of dried adsorbent-free BiSe platelets were dispersed in 18 mL of deionized H_2_O using intense mixing with a magnetic stirrer and simultaneous ultrasonication with a sonicator (VibraCell 505; Sonics & Materials Inc., Newtown, CT, USA). The formed BiSe suspension was added to the acidic CTAB solution. The pH of the formed mixture of CTAB and BiSe platelets was adjusted to pH 2.85 with 0.1 M HBr. Then, a solution of 2.8 mL of TEOS in 5.2 mL of absolute ethanol was added, and the reaction mixture was stirred at room temperature for 90 min. After that time, the pH of the reaction mixture was increased to 5.0 using diluted aqueous ammonia, and the reaction mixture was further stirred at room temperature overnight. Then, the silica-coated nanoparticles were collected using centrifugation for 10 min at 7000× *g*, washed several times with ethanol and water, and finally dispersed in 10 mL of water. The sample was denoted as “**BiSe@S-thin”**.

*Intermediate silica coating:* To obtain an intermediate, 50–80 nm thick silica coating, BiSe@S-thin platelets were used as a starting material. In a flat bottom flask, 10 mL of the BiSe/s-thin aqueous suspension was added and diluted with 40 mL of deionized water. Then, 12 mL of aqueous ammonia (~25%) was added. The mixture was sonicated for 5 min in an ultrasound bath (Sonis 4, Iskra PIO, Šentjernej, Slovenia). After sonication, a solution of 0.7 mL of TEOS in 45 mL of absolute ethanol was added. The formed reaction mixture was further sonicated for 5 min in an ultrasound bath and stirred with a mechanical shaker overnight at ≈200 rpm. Finally, the silica-coated nanoparticles were collected using centrifugation for 10 min at 7000× *g*, washed several times with ethanol and water and dispersed in 10 mL of H_2_O. The sample was denoted as “**BiSe@S-interm**”.

*Thick silica coating*: To obtain a thick, ≈200 nm thick silica coating, BiSe@S-interm platelets were used as a starting material. In a flat bottom flask, 10 mL of the BiSe/s-interm aqueous suspension was diluted with 30 mL of deionized water. Then, 4.8 mL of aqueous ammonia (25%) was added, followed by the addition of a solution of 2.0 mL TEOS in 90 mL absolute ethanol. The formed reaction mixture was sonicated for 5 min in an ultrasound bath and stirred with a mechanical shaker overnight at ≈200 rpm. Finally, the coated nanoparticles were collected using centrifugation for 10 min at 7000× *g*, washed several times with ethanol and water and dispersed in 10 mL of water. The sample was denoted as “**BiSe@S-thick**”.

#### 2.2.3. Synthesis of Silica Spheres

Silica spheres (d ≈ 200 nm) were synthesized using the following procedure. First, 50 mL of deionized water, 40 mL of absolute ethanol, and 9 mL of aqueous ammonia (≈25%) were mixed in a glass flat-bottom flask. Then, a solution containing 3 mL of TEOS and 10 mL of absolute ethanol was added to the glass flask containing water, ethanol, and ammonia. The formed reaction mixture was left to stir on a mechanical shaker overnight. The next day, the formed silica spheres were sedimented using centrifugation at 7000× *g* for 10 min and washed several times with ethanol and deionized water. The silica spheres were finally re-dispersed and stored in deionized water.

#### 2.2.4. Analysis Methods

The synthesized product was characterized using an X-ray powder diffractometer Rigaku MiniFlex (Tokyo, Japan) ( with Cu K*α* radiation (*λ*-1541 Å, 30 kV, 10 mA). For the TEM analysis, the platelets were suspended in ethanol and deposited on a copper-grid-supported lacy carbon film. The TEM analysis was performed using a field-emission electron microscope (JEOL JEM 2100UHR, Tokio, Japan) operating at 200 kV and equipped with an Oxford X-Max80T energy dispersive X-ray spectroscopy detector (EDXS). The width of the platelets expressed as the equivalent diameter was determined from the TEM images, on which 200–300 platelets per sample were utilized for the statistic using Gatan Digital Micrograph Software (Pleasanton, CA, USA). The obtained data, representing the frequency count of the size distribution, were fitted using single or multiple peak Gaussian fit modes. The electrochemical properties (ζ-potential) of the platelets dispersed in water were measured as a function of the suspension pH using a ZetaPALS instrument (Brookhaven Instruments Corporation, Holtsville, NY, USA). The pH of the aqueous suspension was adjusted with diluted hydrochloric acid and sodium hydroxide. 

The light absorption properties were analyzed using classical UV-vis spectroscopy. Spectroscopy was performed with a PerkinElmer Lambda 950 spectrometer (Walham, MA, USA), using a quartz cuvette with a size of 1 × 1 × 3 cm. A measurement range, λ, from 200 to 850 nm was used, with a scanning rate of 1 nm/s. Before the measurements, the water suspension of the platelets was stabilized according to the ζ-potential and sonicated to break any possible agglomerates.

Photo-thermal experiments were performed using an FC-808 Fiber Couple Laser System (CNI Optoelectronics Tech, Changchun, China) configured for continuous-wave operation at 808 nm, with different powers. Laser light was focused on a quartz cuvette with a size of 1 × 1 × 3 cm using an optical lens focusing on a spot size of 8 mm. Control (water) and samples with different concentrations (0.1, 0.3, 0.5, and 1 g/L) were irradiated at a laser power P = 3 W/cm^2^ for 5 min. To test the stability, the samples were cycled 2 times (5 min on, 5 min off). The temperature of the liquid samples was measured with a J-type Teflon thermocouple, which was immersed in the cuvette and connected to a computer to collect the data in real-time. The specific adsorption rate was calculated as follows:$$\mathrm{SAR}=\frac{\rho C_{s}}{m_{sample}}\;{\left(\frac{\Delta T}{\Delta t}\right)}_{t=0}$$
where *ρ* and *C_S_* are the effective density and effective specific heat capacity of the sample, respectively, *m_BiSe_* is the total content of nanoparticles in the sample (g/cm^3^), and ${\left(\frac{\Delta T}{\Delta t}\right)}_{t=0}$ is the slope of the photo-thermal curve linear fit. The Δ*T*/*A* values were calculated by dividing the change of the temperature obtained after 5 min with adsorption measured at 808 nm. In the investigated system, the Bi_2_Se_3_ nanoparticles represent the photo-thermal material. In the case of the uncoated nanoparticles, *m_sample_* = *m*_Bi2Se3_, while in the case of the coated one, *m_sample_* = *m*_Bi2Se3_ + *m_coating_*. Due to this, the concentration of photo-thermal active particles was calculated for each coated sample by calculating the mass share of Bi_2_Se_3_ nanoparticles and the surface layer (silica or EG). For the calculation of the Bi_2_Se_3_ nanoparticle mass, the average particle diameter of 175 nm (size distribution = 50–300 nm), the thickness of 10 nm, and density *ρ* = 6.82 g/cm^3^ were taken. For the silica layer, the average thicknesses, i.e., 3 nm, 65 nm, and 200 nm, for thin, intermediate, and thick silica layers, respectively, and *ρ* = 2 g/cm^3^ was taken. In the case of the solvothermally-synthesized nanoparticles, the correction was made similarly, but taking into account a diameter of nanoparticles of 300 nm, thickness of 10 nm, and EG layer thickness of 2 nm. The ρ of the EG was 1.1 g/cm^3^. 

#### 2.2.5. Hemotoxicity Test

A phosphate-buffered saline (PBS) buffer was prepared by dissolving PBS tablets (Sigma Aldrich, St. Luis, MO, USA) in Milli-Q autoclaved water, to a final concentration of 0.01 M phosphate buffer, 0.0027 M potassium chloride, and 0.137 M sodium chloride. A hemolysis study was performed on the erythrocytes isolated from whole sheep blood. Briefly, whole blood was centrifuged at 800 rpm for 20 min, and the buffy coat was subsequently removed. Erythrocytes were re-dispersed in PBS 1×, 7.4, at 5% *v*/*v* for further analysis. Then, 1 mL of erythrocytes were incubated in triplicate with different concentrations of each nanoparticle sample for 3 h at 37 °C with constant orbital shaking. After incubation, tubes were centrifuged (1500 rpm/4 min) to sediment cells, and the supernatant was analyzed in triplicates. The hemolysis was evaluated by measuring the absorbance of the released hemoglobin (*A*) at 541 nm in the supernatant using a spectrophotometer (BioTek Synergy H4 Hybrid microplate reader, Winooski, VT, USA). Each sample was measured in triplicate. Samples representing “100% dead” were prepared by lysing control samples with deionized water via hypotonic osmotic shock, while samples representing “0%” as a negative control were simply incubated with buffer. The hemolysis was then calculated as follows: *Hemolysis (%) = 100 (A_sample_ − A_control_)/(A_100% dead_ − A_control_)*. Mixed-effect ANOVA (Graphpad 8.1, Prism) was used to test whether the nanoparticle-correlated hemolysis was significantly higher than the negative control. Data are presented as the mean ± standard deviation (SD) for all experiments.

#### 2.2.6. In Vitro Cell Viability Assay

HeLa cells (ATCC CCL-2) were grown at 37 °C in a humidified atmosphere and 5% CO_2_. The growth medium was DMEM, supplemented with 10% FBS, 100 μg/mL penicillin, and 100 μg/m streptomycin. The medium was changed every 2–3 days. When the cells reached 70–90% confluence, they were detached with a 0.25% trypsin solution and subculture at a 1:5 ratio. Nanoparticles (2 g/L) in PBS were diluted in the growth medium, to obtain final concentrations of 0.1, 0.3, 0.5, and 1 g/L. The viability of the samples exposed to different concentrations of nanoparticles was compared with pure PBS diluted in the same ratio with growth medium. Untreated (mock-treated) cells served as a negative control for the viability calculations (value 1). HeLa cells were seeded at 2 × 10^4^ cells/well in a 96-well microtiter plate and grown overnight. Cells were then exposed to nanoparticles or PBS in the growth medium. After incubation in a cell incubator for 24 h, the cells were washed with PBS and incubated with 10% PrestoBlue Viability Reagent in a growth medium for 1.5 h. The viability of HeLa cells was determined by measuring the metabolic conversion of the PrestoBlue resazurin-based dye to highly fluorescent resorufin using an Infinite F200 plate reader (Tecan, Grödig, Austria) at 560 nm excitation and 595 nm emission. The fluorescence intensities obtained were corrected for background fluorescence and normalized to mock-treated HeLa cells. Three independent experiments, each with three technical replicates, were performed to evaluate the viability of the HeLa cells. Cells were analyzed using two-way ANOVA. Dunnet’s multiple comparisons test was used to compare cells exposed to PBS only with cells exposed to different nanoparticles at the same concentrations. Tukey’s multiple comparisons test was used to compare different concentrations within each treatment. Multiplicity-adjusted P values were calculated for each comparison. Statistical analyses of the in vitro toxicity assays were performed using GraphPad Prism 8 software (GraphPad Software, San Diego, CA, USA).

## 3. Results and Discussion

To investigate the possibility of using silica coating as an alternative to EG coating, the hydrothermally synthesized, adsorbent-free Bi_2_Se_3_ nanoparticles were coated with silica layers of three different thicknesses. To evaluate the effect of the surface silica layer, the coated nanoparticles were compared with their solvothermally-prepared (EG-coated) counterparts [63]. 

Figure 1 shows the XRD patterns of the hydrothermally- and solvothermally-synthesized Bi_2_Se_3_ nanoparticles (BiSe and BiSe-ST, respectively). The diffraction patterns are very similar. The peaks in both cases can be indexed according to the rhombohedral structure of Bi_2_Se_3_ (space group *R3m*, *JCPDS 33-0214).* The strong and sharp diffraction peaks indicate that the Bi_2_Se_3_ nanoparticles were well crystallized. The TEM analysis of BiSe and BiSe-ST nanoparticles showed that the nanoparticles had a hexagonal plate-like morphology. As was expected according to theory [65], the size distribution of the hydrothermally synthesized nanoparticles was broader (majority with d ≈ 50–300 nm, a minority with d > 300 nm) compared to the size distribution of their BiSe-ST counterparts (d ≈ 280–380 nm) [7]. EDXS analysis showed that the platelets had an atomic ratio Bi/Se = 0.66 ± 0.1, which was in good agreement with Bi_2_Se_3_ stoichiometry. As was shown in our previous study, the most important difference between the BiSe and BiSe-ST nanoparticles was only observed with HR-TEM, when the nanoparticles were orientated edge-on, with their large surface parallel to the electron beam. While the BiSe nanoparticles were clean, the BiSe-ST nanoparticles were coated with a thin, approximately 2 nm thick, ethylene-glycol layer (see Ref. [7] for more details).

Figure 2 shows the BiSe nanoparticles subsequently coated with (a) a thin ≈ 2–5 nm, (b) an intermediate ≈ 50–80 nm, and (c) a thick ≈ 200 nm uniform silica layer covering the individual nanoparticles. An additional confirmation that the silica coating was successful was the change in the point of zero charges (PZC). Figure 2d shows the zeta potential (ζ) of the silica-coated BiSe nanoparticles compared to the uncoated ones and silica nanospheres synthesized using a modified Stober process [64]. The PZC decreased from pH = 5 (BiSe) to ≈2.5 for BiSe@S-thin and ≈3 for BiSe@S-thick.

In our previous work [7], we demonstrated that an EG layer, adsorbed on the surface of the Bi_2_Se_3_ nanoparticles, can mask/impair the optical properties derived from TSS. Namely, BiSe-ST nanoparticles display absorption peaks in the range of ≈200–350 nm and ≈350–550 nm. Near the IR region, the absorbance decreases without showing absorption peaks [7]. On the contrary, the absorption spectrum of BiSe nanoparticles in the spectral range of ≈200–300 nm is very similar to that of BiSe-ST. However, a striking difference can be observed at higher wavelengths, where BiSe nanoparticles display an intense and broad adsorption peak over the entire measured range. This broad peak can be ascribed to LSPR, which results from TSS [5]. Figure 3 shows the normalized UV-Vis absorption spectra of the silica-coated Bi_2_Se_3_ nanoparticles compared to their BiSe and BiSe-ST counterparts. Normalization of UV-Vis spectra was used to qualitatively evaluate the impact of the coating layer on the optical properties of the Bi_2_Se_3_ nanoparticles. The BiSe@S-thin and BiSe@S-interm samples displayed similar absorption spectra to the BiSe nanoparticles over the entire measuring area. They showed absorption peaks in the spectral range ≈ 210–300 nm, where both BiSe and pure silica spheres adsorb. At a higher wavelength, they showed a broad adsorption peak, dominating over the whole measured spectral region from ≈380 to 850 nm, which can be ascribed to the LSPR resulting from TSS. The only difference in the adsorption of BiSe@S-thin and BiSe@S-interm compared to the BiSe sample was a less pronounced absorption peak in this region, which can be attributed to the silica coating. However, the results suggest that the thin (≈2–5 nm) and intermediate (≈50–80 nm) silica layers did not mask the optical properties derived from TSS. With the increase in the silica layer thickness, the absorption spectrum also changed. In the spectral range from ≈200–350 nm, the BiSe@S-thick sample displayed absorption spectra very similar to the BiSe, BiSe@S-thin, and BiSe@S-inter samples. Towards higher wavelengths, the absorption of the BiSe@S-thick sample decreased, similarly to the case of the BiSe-ST sample.

According to theory [66], nanoparticles that display LSPR also display a photo-thermal effect, i.e., the effect in which a material can convert light into heat [67]. This effect makes TI nanoparticles relevant for biomedical applications, which has been reported previously [6,8,9,10,11,12,13,14]. Figure 4a shows the photo-thermal performance of the silica-coated Bi_2_Se_3_ nanoparticles suspended in water at a concentration of 0.1 g of particles/L. The results were compared with the performance of the BiSe and BiSe-ST counterparts. The measurements show that the BiSe@S-thin and BiSe@S-interm samples exhibited better photo-thermal conversion, while the sample BiSe@S-thick exhibited a similar conversion to the BiSe sample. Compared to the solvothermally-prepared nanoparticles (BiSe-ST), the silica-coated nanoparticles showed a much lower photo-thermal conversion. The recovery after the 1st cycle was ≥94% for the uncoated and the silica-coated nanoparticles, while the recovery of the BiSe-ST samples was only ≈ 83% (Figure 4b). The photo-thermal conversion of the nanoparticles increased linearly with increasing suspension concentration (Figure 4c). The final temperature obtained after 5 min for each sample and the concentration was measured in triplicate. The difference was less than 0.5 °C, which could be attributed to a systematic error of the thermocouple. The determined specific adsorption rate (SAR) showed that the silica-coated nanoparticles had a significantly higher SAR compared to their BiSe counterparts (≈760 W/g). The SAR values of the silica-coated nanoparticles decreased when increasing the thickness of the silica layer (≈1150 W/g, ≈1080 W/g, ≈940 W/g for BiSe@S-thin, BiSe@S-interm and BiSe@S-thick, respectively). However, the highest SAR was determined for the BiSe ST sample (≈2400 W/g) (Figure 4d). To date, there have been no reports of the SAR of Bi_s_Se_3_ nanoparticles, which is a very important parameter for the determination of the quality of photo-thermal materials. It describes the ability of a material to adsorb radiation per mass unit. Compared to the previous photo-thermal material, Au-nanorods (≈10 kW/g) [67], the coated Bi_2_Se_3_ nanoparticles displayed a moderate SAR. Another parameter describing the quality of photo-thermal materials is ΔT/A, which describes the efficiency of a material in converting adsorbed light into heat (Figure 4e). BiSe ST, the sample with the highest SAR, displayed the lowest conversion efficiency, which was even lower than BiSe. In contrast, the silica-coated nanoparticles showed higher ΔT/A values compared to the BiSe and BiSe ST samples.

When determining photo-thermal properties, the concentration/mass of the photo-thermal particles that actually perform the photo-thermal conversion is essential. In our system, the Bi_2_Se_3_ nanoparticles represent the photo-thermal part of the material that absorbs light and converts it into heat. For the uncoated nanoparticles (BiSe), the mass of the photo-thermal active material was equal to the overall nanoparticle mass. In the case of coated nanoparticles, the mass of the nanomaterials is the sum of the photo-thermal core and the mass of the coating, either EG or silica. Taking this into account, the photo-thermal conversions shown in Figure 4a,c do not represent the true photo-thermal values of Bi_2_Se_3_ nanoparticles alone, but of the Bi_2_Se_3_ and silica coating together. Namely, at a certain concentration, the amount of the photo-thermal active nanoparticles for the BiSe sample was equal to the mass of the sample, while in the case of the coated BiSe-ST, BiSe@S-thin, BiSe@S-interm, and BiSe@S-thick samples, the mass of the photo-thermal active particles was ≈3.6%, ≈26.6%, ≈93%, and ≈99.1% lower, respectively, which was due to the coating (more details in the experimental section). To show the actual photo-thermal conversion of the Bi_2_Se_3_ nanoparticles, the final temperatures obtained after 5 min had to be plotted as the temperature at a certain concentration of Bi_2_Se_3_ nanoparticles in the sample (Figure 4f). Here, we see that the silica coating enhanced the photo-thermal conversion. The conversion of BiSe@S-thin was slightly improved compared to BiSe but was lower compared to BiSe-ST. When increasing the silica layer, the photo-thermal conversion increased. The BiSe@Si-interm and BiSe@Si-thick samples displayed significantly higher conversion compared to the BiSe, BiSe-ST, and BiSe@S-thin samples. The desired temperatures could be achieved with a lower concentration of photo-thermal active nanoparticles. For instance, a temperature in the range of 43–45 °C (temperature needed for the cell apoptosis [68,69,70,71]) was reached with significantly lower concentrations of photo-thermal active nanoparticles than in the case of the BiSe, BiSe-ST, and BiSe@S-thin samples.

This increase in the photo-thermal conversion of the silica-coated nanoparticles compared to the uncoated nanoparticles is consistent with reports in the literature. Namely, the silica shell was transparent to visible and near-IR radiation. In addition to its protective role, the silica coating also improved the thermodynamic stability of the nanoparticles in suspension, and by suppressing light reflection, it improved the light absorption of nanoparticles [72,73]. The same can also be concluded for the EG-coated nanoparticles. The high photo-thermal conversion (>94%) obtained over multiple cycles and sufficient SAR values make silica-coated nanoparticles good candidates for photo-thermal mediators in biomedical applications.

Nanoparticles for biomedical applications must be non-toxic. Figure 5a shows the hemotoxicity of the samples. The mixed-effect ANOVA test, which compared samples with the PBS (negative control), showed that the hemolysis effect was significantly higher for BiSe@S-interm at 0.3 and 0.5 g/L. However, at a concentration of 1 mg/mL for the same sample, the hemolysis was found to be 0.3%. Therefore, this discrepancy falls within the range of experimental method error, and we can conclude that the sample did not cause significant hemolysis. In all cases, the determined hemolysis was lower than 2%, which falls within the error area of the method and indicates that neither the BiSe and BiSe-ST nor silica coated Bi_2_Se_3_ nanoparticles were hemotoxic under the given conditions.

The biocompatibility of the silica-coated nanoparticles was further tested on a more complex biological system of human cell lines. Figure 5b shows the viability of the HeLa cells after 24 h of exposure to the nanoparticles, with test concentrations ranging from 0.1 to 1 g/L. No significant decrease in cell viability was observed at the lowest concentration of 0.1 g/L compared with PBS for all tested materials (two-way ANOVA, Dunnett’s multiple comparisons test). However, at higher concentrations (0.3–1 g/L), the viability of cells exposed to EG-coated nanoparticles was significantly reduced, whereas no significant decrease in cell viability was observed with the pristine or silica-coated Bi_2_Se_3_ nanoparticles. The viability of cells exposed to BiSe, BiSe@S-thin, and BiSe@S-interm decreased slightly at concentrations higher than 0.5 g/L, whereas BiSe@S-thick showed a similar cell viability to the control, even at higher concentrations. In contrast, the BiSe ST nanoparticle cell viability decreased to 80% at 0.1 g/L, and this effect was even more pronounced at higher concentrations (20%, 10%, and ≈0% cell viability for 0.3, 0.5, and 1 g/L, respectively).

To date, there have been few reports of the use of EG-coated Bi_2_Se_3_ nanoparticles as photo-thermal mediators for potential biomedical applications [9,10,11,12]. No cytotoxicity was observed at lower concentrations, but with concentrations ≥1 g/L, the viability was significantly reduced [10,11], which was similar to our results. The in vitro assay on HeLa cells confirmed our predictions and is consistent with the reported limitations, whereby EG-coated nanoparticles are toxic and, therefore, not appropriate for biomedical applications. On the other hand, the silica-coated Bi_2_Se_3_ nanoparticles showed an almost negligible cytotoxic effect and are therefore more suitable for biomedical applications.

## 4. Conclusions

LSPR is responsible for the occurrence of a photo-thermal effect in Bi_2_Se_3_ TI nanoparticles, which makes them relevant in the field of medical diagnostics and therapy. However, a problem that limits the application of Bi_2_Se_3_ in the mentioned field is colloidal instability and oxidation in the physiological medium [13]. Therefore, the nanoparticles have to be coated with a protective surface layer. However, in the case of the TI, a protective layer can alter or mask the surface and optical properties originating from TSS, as was demonstrated in the case of an EG coating [7]. 

There have been reports where EG coated Bi_2_Se_3_ nanoparticles were used as photo-thermal mediators in biomedical applications [6,8,9,10,11,12,13,14]. However, as shown in this work, EG coated nanoparticles are cytotoxic and, therefore, not suitable for clinical applications. 

The alternative is a protective surface layer made of silica, which, unlike EG, is FDA and EFCA-approved, being non-toxic and biodegradable. As we demonstrated here, silica coated Bi_2_Se_3_ nanoparticles with thin (≈2–5 nm) and intermediate (≈50–80 nm) thick silica layer do not alter or mask the optical properties that originate from TSS, as is observed in the case of their EG coated counterparts. A change in the optical properties only appears in the case of the thickest (≈200 nm) silica layers. Here, a thick silica layer probably masks the true optical properties of the TI. 

Compared to the uncoated and EG coated Bi_2_Se_3_ nanoparticles, the silica coated nanoparticles displayed an improved photo thermal effect, which increased with an increasing silica layer. The desired temperatures could be reached with 10–100 times lower concentrations of photo-thermal active nanoparticles compared to their uncoated and EG coated counterparts. With in vitro tests performed on erythrocytes and HeLa cells, we showed that, unlike EG coated nanoparticles, the silica coated nanoparticles are biocompatible. Silica coated Bi_2_Se_3_ nanoparticles displayed negligible hemolysis and cytotoxicity, even at high concentrations (>0.5 g/L). Nevertheless, although uncoated Bi_2_Se_3_ nanoparticles are biocompatible, however, they cannot be used in biomedical applications, since they are not colloidally and chemically stable in the physiological medium.

An improved colloidal stability, easy functionalization, high photo-thermal conversion, moderate SAR, and biocompatibility show that silica is a suitable alternative for coating and, therefore, such silica-coated Bi_2_Se_3_ nanoparticles are a suitable candidate photo-thermal material for biomedical applications.

## Figures and Tables

**Figure 1 nanomaterials-13-00809-f001:**
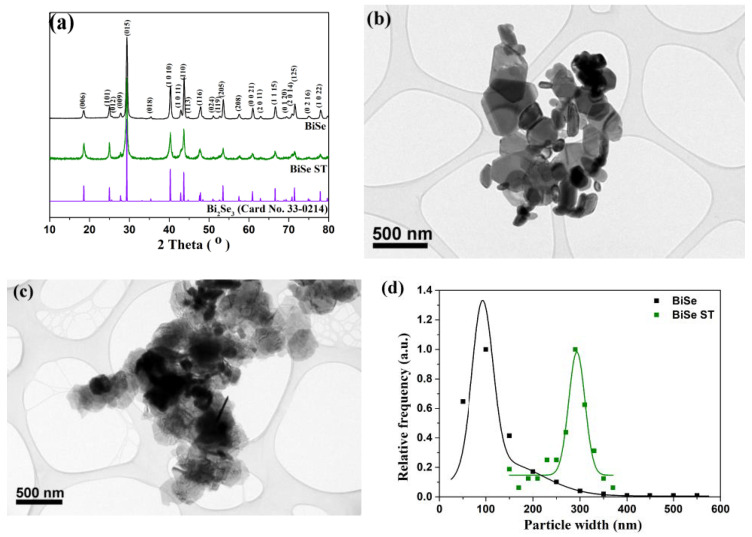
XRD patterns of Bi_2_Se_3_ nanoparticles synthesized hydrothermally (BiSe) and solvothermally (BiSe ST) (**a**). Representative TEM image of BiSe (**b**) and BiSe-ST (**c**) nanoparticles, and the corresponding size distribution (**d**), obtained from TEM images.

**Figure 2 nanomaterials-13-00809-f002:**
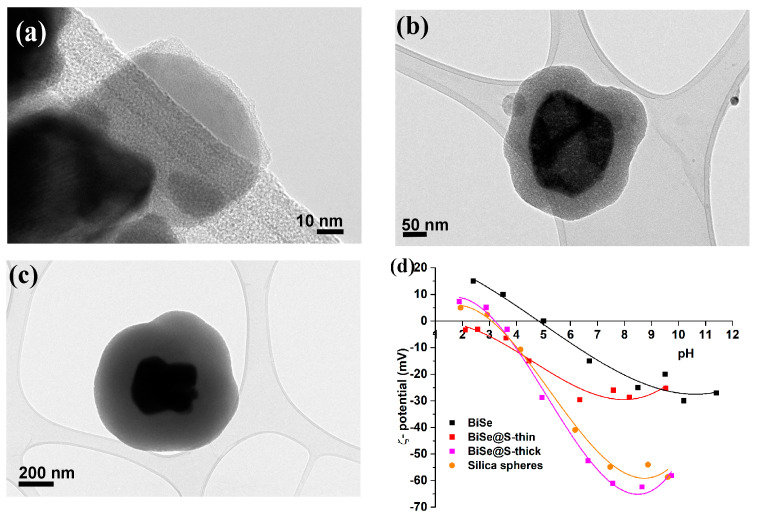
TEM image of silica-coated Bi_2_Se_3_ nanoparticles with thin ≈ 2–5 nm (**a**), intermediate ≈ 50–80 nm (**b**) and thick ≈ 200 nm silica layer (**c**). Zeta-potential behavior of silica-coated Bi_2_Se_3_ nanoparticles compared to uncoated nanoparticles and silica nanospheres (**d**).

**Figure 3 nanomaterials-13-00809-f003:**
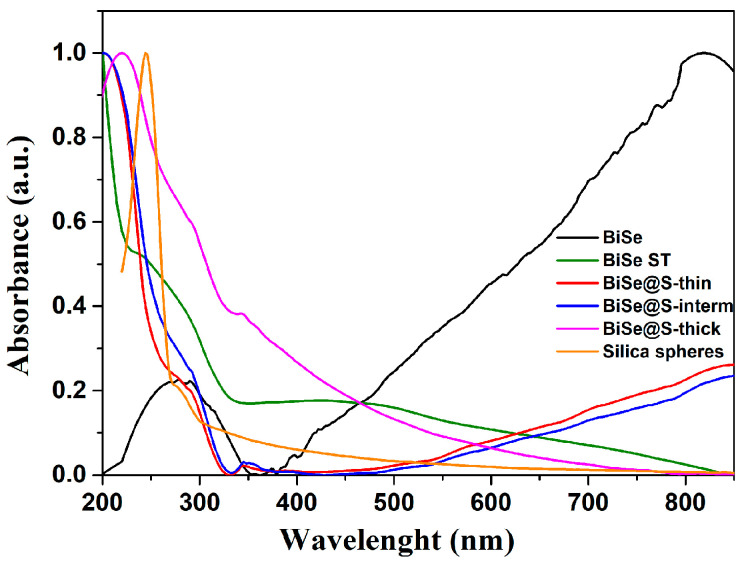
Normalized UV-Vis absorption spectra of the silica-coated nanoparticles with the thin (≈2–5 nm), intermediate (≈50–80 nm), and thick (≈200 nm) silica layer in comparison with the clean, EG coated nanoparticles, and silica spheres. The measurement was done on a suspension with a concentration of 0.1 g/L.

**Figure 4 nanomaterials-13-00809-f004:**
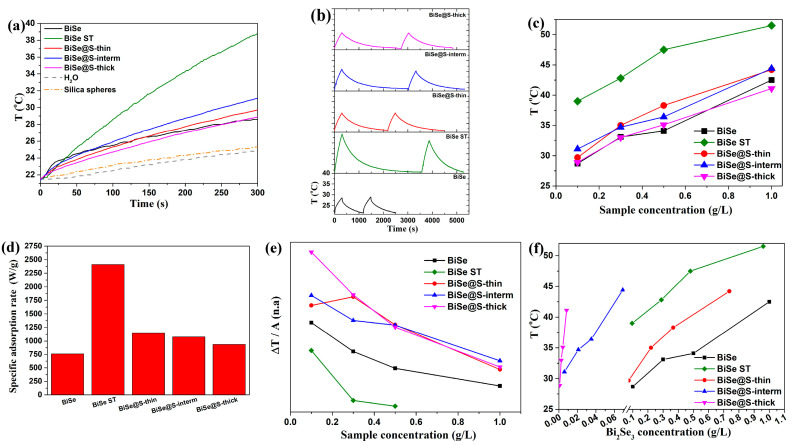
Temperature increase of the particle solution with 0.1 g/L (**a**) and corresponding heating and cooling cycles (**b**). Final T reached after 5 min laser exposure at different concentrations (**c**), specific adsorption rate (SAR) (**d**) and efficiency of adsorbed light conversion into heat represented as ΔT/A for different concentrations (**e**). Final temperatures after 5 min laser exposure vs. concentration of the photo-thermal active particles in the sample (**f**). The experiments were conducted with laser light of 808 nm and P = 3 W/cm^2^.

**Figure 5 nanomaterials-13-00809-f005:**
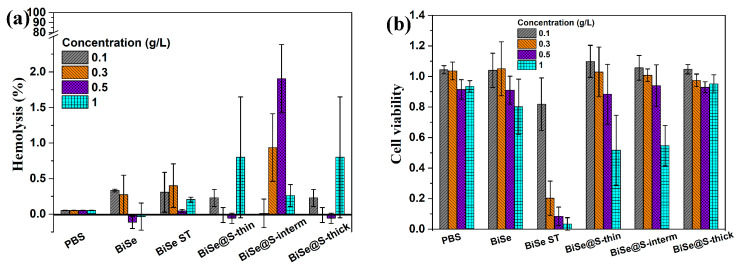
Influence of silica coated Bi_2_Se_3_ nanoparticles compared to their pristine and ethylene glycol coated counterparts on red blood cells toxicity (**a**) and HeLa cell viability (**b**). Cells were exposed to different concentrations of nanoparticles and compared with the control (PBS medium). Negative control for hemotoxicity was 0%, and control for HeLa cell viability was 1.

## Data Availability

The data presented in this study are available on request from the corresponding author.

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
