# Peer review of "Silica Coated Bi_2_Se_3_ Topological Insulator Nanoparticles: An Alternative Route to Retain Their Optical Properties and Make Them Biocompatible"

_nanomaterials, 2023, doi:10.3390/nano13050809_

Round 1
Reviewer 1 Report
The manuscript described the synthesis of Bi2Se3 nanoparticles. The manuscript is well-written and has sufficient data. However, many issues must be improved before publication.
The title is strange which was used “;”? Is it “:”? The authors should check and correct this point.
Since the manuscript described many steps in the synthesis of nanoparticles, the author should supply a scheme to describe the overall synthesis process and application of the introduced nanoparticles. There is no overall image that described the experiment for the synthesis NPs as well as the design experiment. Therefore, this kind of schematic figure would help for improving the quality of the manuscript as well as eye-catching for the readers.
There are a lot of typos in the manuscript (e.g. lines 406, line 409, the semicolon in the title should be colon; “Bi2Se3” in the title and abstract should be Bi2Se3. “In-vitro” should be In vitro; Between “×” and word should have space; there should be a space between the number and unit; The format of citation 10 and 11 in the introduction is wrong…)
The author should provide statistical analysis to demonstrate the reproducibility of the research.
How many times the experiments were repeated? This information should be added where are appropriate.
For the Hela cell experiment, the authors are suggested to add some real images which validate live/dead cell tests to support the non-toxicity of the cell during the culture process. Also, how long the Hela cell was culture? Which is the concentration of NPs not affecting the cell growth or maximum concentration not causing cell death?
The manuscript described the synthesis of Bi2Se3 nanoparticles. The manuscript is well-written and has sufficient data. However, many issues must be improved before publication.
The title is strange which was used “;”? Is it “:”? The authors should check and correct this point.
Since the manuscript described many steps in the synthesis of nanoparticles, the author should supply a scheme to describe the overall synthesis process and application of the introduced nanoparticles. There is no overall image that described the experiment for the synthesis NPs as well as the design experiment. Therefore, this kind of schematic figure would help for improving the quality of the manuscript as well as eye-catching for the readers.
There are a lot of typos in the manuscript (e.g. lines 406, line 409, the semicolon in the title should be colon; “Bi2Se3” in the title and abstract should be Bi2Se3. “In-vitro” should be In vitro; Between “×” and word should have space; there should be a space between the number and unit; The format of citation 10 and 11 in the introduction is wrong…)
The author should provide statistical analysis to demonstrate the reproducibility of the research.
How many times the experiments were repeated? This information should be added where are appropriate.
For the Hela cell experiment, the authors are suggested to add some real images which validate live/dead cell tests to support the non-toxicity of the cell during the culture process. Also, how long the Hela cell was culture? Which is the concentration of NPs not affecting the cell growth or maximum concentration not causing cell death?
Author Response
Dear Editor
We appreciate the reviewer's valuable comments on our paper. We have implemented them as addressed below and highlighted in the revised manuscript. Since there were some concerns regarding the language, we checked the manuscript again by the licensed version of the Grammarly programme and by a native speaker from the field. However, besides typing mistakes, no other major mistakes related to grammar were found.
Reviewer 1: The manuscript described the synthesis of Bi2Se3 nanoparticles. The manuscript is well-written and has sufficient data. However, many issues must be improved before publication.
COMMENT 1: The title is strange which was used “;”? Is it “:”? The authors should check and correct this point.
Response:
- After additional inspection regarding the meaning of “;” and “:”and consolidation with the native speaker, we agree with the reviewer that “:” should be used. We corrected this in the manuscript.
COMMENT 2: Since the manuscript described many steps in the synthesis of nanoparticles, the author should supply a scheme to describe the overall synthesis process and application of the introduced nanoparticles. There is no overall image that described the experiment for the synthesis NPs as well as the design experiment. Therefore, this kind of schematic figure would help for improving the quality of the manuscript as well as eye-catching for the readers.
Response:
- Thank you for the suggestion. However, we think that the synthesis of Bi2Se3 nanoparticles, their coating and final characterisation are represented and well described in the Experimental section. Therefore, there is no need for an additional scheme. We divided separately the synthesis of clean (hydrothermally synthesised, described in lines 132-139) and ethylene glycol coated (solvothermally synthesised, described in lines140-145) Bi2Se3
After that, the step-by-step silica coating process is described in a separate paragraph for each layer thickness (thin, intermediate in lines 164-160, 161-171 and 172-181, respectively).
Additionally, for more details regarding the nanoparticles’ synthesis and silica coating process, we also added relevant references.
Finally, in a separate paragraph, their characterisation (lines 192-238), hemolysis (239-258) and in vitro test were described (259-283).
COMMENT 3: There are a lot of typos in the manuscript (e.g. lines 406, line 409, the semicolon in the title should be colon; “Bi2Se3” in the title and abstract should be Bi2Se3. “In-vitro” should be In vitro; Between “×” and word should have space; there should be a space between the number and unit; The format of citation 10 and 11 in the introduction is wrong…).
Response:
- The semicolon was changed to the colon
- Bi2Se3 in the title and abstract was corrected to Bi2Se3 (highlighted in the revised manuscript)
- In-vitro was corrected to in vitro as suggested (highlighted in the revised manuscript)
- Typing mistakes regarding the spaces were corrected.
- Lines 406 and 409: The authors did not find the typo in lines 406 and 409. The only problematic word in this line is “hemotoxicity” and “hemolysis”. There can be two variations, the one which is written in the manuscript or “heamotoxicity” and “heamolysis”. By consulting with our native speaker, both versions can be used.
- The format in the case of references 10 and 11 was corrected. The correction is highlighted in the revised manuscript.
- The manuscript was additionally read and checked by a native speaker.
COMMENT 3: The author should provide statistical analysis to demonstrate the reproducibility of the research.
- For the XRD, TEM, SEM, UV-VIS and ζ-potential analysis, only one measurement per sample was done. The results are similar to those in previously reported research 1,2. Slight differences in the results (> 1% of an error) can be attributed to systematic errors (method, equipment, ...).
- The width of the nanoparticle expressed as an equivalent diameter was determined from TEM images, where 200-300 nanoparticles per sample were taken for the statistic. The obtained data, representing the frequency count of the size distribution, were fitted using the single or multiple peak Gaussian fit mode. The size distribution is presented in Figure 1d. The detailed description is included in the Experimental section in lines 198-202.
- For the photo-thermal effect analysis, the replicates were made only in the case of determining the temperatures as a function of nanoparticles concentration (Figure 5 c). Each concentration and sample was measured three times (triplicates). However, the temperature difference between the replicates was minimal.
We added the explanation, which is highlighted in the text (lines 365-367): The final temperature obtained after 5 minutes for each sample and concentration was measured in triplicates. The difference was less than 0.5 °C, which can be attributed to the systematic error of the thermocouple.
- As described in the Experimental part, for the hemolysis, each concentration was prepared and measured in triplicates. This information is given in line 251.
- The information about the in vitro test on HeLa cells and statistical analysis was added in the paragraph on In vitro cell (highlighted lines 276-283).
Three independent experiments, each with three technical replicates, were performed to evaluate the viability of HeLa cells. Cells were analysed by two-way ANOVA. Dunnet's multiple comparisons test was used to compare cells exposed to PBS only with cells exposed to different nanoparticles at the same concentrations. Tukey's multiple comparisons test was used to compare different concentrations within each treatment. Multiplicity-adjusted P values were calculated for each comparison. Statistical analyses of the in vitro toxicity assays were performed using GraphPad Prism 8 software (GraphPad Software, CA, USA).
COMMENT 4: How many times the experiments were repeated? This information should be added where are appropriate.
Response: This is explained in reply to Comment 3.
COMMENT 5: For the Hela cell experiment, the authors are suggested to add some real images which validate live/dead cell tests to support the non-toxicity of the cell during the culture process. Also, how long the Hela cell was culture? Which is the concentration of NPs not affecting the cell growth or maximum concentration not causing cell death?
Response:
- The viability of the HeLa cells and therefore determining the cytotoxicity was performed by measuring the metabolic conversion of the PrestoBlue resazurin-based dye by highly fluorescent resorufin on the plate reader. The metabolic conversion was detected by measuring the fluorescence intensity. This method is one of the standard methods for determining cell viability. The measurements were not performed on the fluorescent microscope.
- Additional information about the growth of HeLa cells is provided in the Experimental section. The added part (lines 268-276 in the revised manuscript) is highlighted.
HeLa cells were seeded at 2 x 104 cells/well in a 96-well microtiter plate and grown overnight. Cells were then exposed to nanoparticles or PBS in the growth medium. After incubation in a cell incubator for 24 hours, the cells were washed with PBS and incubated with 10 % PrestoBlue Viability Reagent in a growth medium for 1.5 hours. The viability of HeLa cells was determined by measuring the metabolic conversion of the PrestoBlue resazurin-based dye to highly fluorescent resorufin using the Infinite F200 plate reader (Tecan, Grödig, Austria) at 560 nm excitation and 595 nm emission. The fluorescence intensities obtained were corrected for background fluorescence and normalised to mock-treated HeLa cells.
- As presented in Figure 5b and discussed in the text, in the case of the ethylene-glycol coated nanoparticles, a 20 % viability decrease was observed already at the lowest investigated concentration (0.1 g/L). For the silica-coated Bi2Se3 nanoparticles, the cell viability slightly decreases at concentrations larger than 0.5 g/L. However, as discussed in the manuscript, the same trend was observed for the control sample. According to the literature, the cell viability for ethylene-glycol coated Bi2Se3 nanoparticles already decreased at 0.02 g/L 3,4. The concentrations can vary slightly depending on the size of the nanoparticles and the thickness of the layer.
References
(1) Belec, B.; Ferfolja, K.; Goršak, T.; Kostevšek, N.; Gardonio, S.; Fanetti, M.; Valant, M. Sci. Rep. 2019, 9, 190571.
(2) Kralj, S.; Makovec, D. ACS Nano 2015, 9, 9700.
(3) Li, Z.; Liu, J.; Hu, Y.; Howard, K. A.; Li, Z.; Fan, X.; Chang, M.; Sun, Y.; Besenbacher, F.; Chen, C.; Yu, M. ACS Nano 2016, 10, 9646.
(4) Li, J.; Jiang, F.; Yang, B.; Song, X. R.; Liu, Y.; Yang, H. H.; Cao, D. R.; Shi, W. R.; Chen, G. N. Sci. Rep. 2013, 3, 1998.
Reviewer 2 Report
The authors coated Bi2Se3 nanoparticles with silica oxide. The coating thickness and biocompatibility were tested. This is an interesting paper which could promote the applications of bismuth selenide nanoparticles.
I recommend the publication after proper modification. The detailed comments were listed as follows:
1) Is there any leak from the core of the nanoparticles after the coating? I think it should be significant between the different thicknesses of silica layer. The data would help the readers to understand the applications.
2) Is there any possibility of repeat using of the coating nanoparticles? This could be added to the revised manuscript.
3) The English expression could be improved.
Author Response
Reviewer 2: The authors coated Bi2Se3 nanoparticles with silica oxide. The coating thickness and biocompatibility were tested. This is an interesting paper which could promote the applications of bismuth selenide nanoparticles.
I recommend the publication after proper modification. The detailed comments were listed as follows:
COMMENT 1: Is there any leak from the core of the nanoparticles after the coating? I think it should be significant between the different thicknesses of silica layer. The data would help the readers to understand the applications.
Response: Thank you for the relevant question. In the framework of this research, we did not investigate the possible leakage from the core nanoparticles. This investigation was as base research, which will help us to choose the best-performing nanoparticles for further investigation. We are aware that the possible leakage needs be investigated in the future, if such nanoparticles would be used in applications. Since there are no reports in the literature, we only investigated if the silica coating is appropriate to substitute for EG or PEG and how layer thickness affects the optical and photo-thermal properties. As described in the manuscript, in the case of the ethylene-glycol coated nanoparticles, optical properties resulting from topological states are masked.
COMMENT 2: Is there any possibility of repeat using of the coating nanoparticles? This could be added to the revised manuscript.
Response: As it is presented in the Figure 5b, the silica coated nanoparticles display high recovery level (> 94%). This means, that after first cycle, they retain almost same photo-thermal properties and therefore, they can be reused again. The authors think, that by presenting the recovery cycles on Figure 5b and by reporting the recovery % is enough that the reader can understand that they can be used multiple times. Additionally, we added that the silica coated nanoparticles can be reused. This was emphasised in the revised manuscript, line 414.
COMMENT 3: The English expression could be improved.
Response: The manuscript was again revised and corrected by the Licensed Grammarly program and native speaker.
Reviewer 3 Report
In this manuscript, Dr. B. Belec and co-workers propose to apply a coating of silica to Bi2Se3 nanoparticles as to prevent their aggregation in aqueous physiological media while preserving the optical properties; the silica coating should help to overcome the drawback of EG, that is toxic for cells, i.e. not suited for biomedicine applications.
This study is interesting for the readership specialised in the field however, some issues should be solved before publication.
1) First of all, I am quite mesmerised by "nanoparticles" sizes, especially for the "thick layer of silica" case. In fact, as also shown by the TEM image reported in Figure 2c, such particles have a diameter of more than 800nm. This is hardly "nano"...maybe the authors should briefly point out that such nanoparticles are "out of size". Such very large size affects the optical properties, as shown in Figure 3, where the plasmon in the rose spectrum is hardly observable (a very small shoulder on the right of the silica absorption).
2) As for Figure 3: why does the absorption peak related with silica shift? is it still a plasmonic absorption that varies with particles size? this is not clarified in the text.
3) At page 9 the authors discuss the phototermal properties of NPs, and extensively compre photo-termal conversion behavior for the three silica-coated systems and standard however, a comparison with the EG-coated Bi2Se3 nanoparticles is neither reported or discussed. The authors should at least mention some comparison based on literature.
4) Finally, in the conclusions section (line 453) the authors affirm that "as it is shown in this work, EG coated nanoparticles are cytotoxic...". But at page 10-11, where biocompatibility studies are reported, no tests are performed on EG-coated nanoparticles. There is a reference (46) mentioned at line 433, but this is a literature study, not a finding arising by this work, as claimed in the conclusions.
Author Response
Reviewer 3: In this manuscript, Dr. B. Belec and co-workers propose to apply a coating of silica to Bi2Se3 nanoparticles as to prevent their aggregation in aqueous physiological media while preserving the optical properties; the silica coating should help to overcome the drawback of EG, that is toxic for cells, i.e. not suited for biomedicine applications.
This study is interesting for the readership specialised in the field however, some issues should be solved before publication.
COMMENT 1: First of all, I am quite mesmerised by "nanoparticles" sizes, especially for the "thick layer of silica" case. In fact, as also shown by the TEM image reported in Figure 2c, such particles have a diameter of more than 800nm. This is hardly "nano"...maybe the authors should briefly point out that such nanoparticles are "out of size". Such very large size affects the optical properties, as shown in Figure 3, where the plasmon in the rose spectrum is hardly observable (a very small shoulder on the right of the silica absorption).
Response: We agree with the reviewers’ observation that nanoparticles (Bi2Se3 coated with silica) are a little bit “out of size” and can barely be named s nanoparticles. However, the term nanoparticle is referred to the core nanoparticles (Bi2Se3) and not the “composite”. For this, we use a description such as Bi2Se3 coated with a silica layer.
Regarding the optical properties: as stressed in the introduction, the layer deposited on the topological insulator nanoparticles can alter their optical properties, especially in the range from ≈ 400 nm above. In this spectral range, the Bi2Se3 nanoparticles display localised surface plasmon resonance (LSPR) due to the topological surface states (property which defines the topological insulators). The in deep investigation of optical properties is presented in our previous manuscript1. In this research, we investigated the effect of the silica layer thickness on the optical properties of the topological insulator resulting from TSS. We tracked the original optical properties of the uncoated Bi2Se3 nanoparticles in the range where LSPR features appear and how the silica layer thickness affects them. Namely, if the optical properties resulting from TSS are still present or they are masked, as in the case of the ethylene-glycol coated nanoparticles (BiSe-ST).
COMMENT 2: As for Figure 3: why does the absorption peak related with silica shift? is it still a plasmonic absorption that varies with particles size? this is not clarified in the text.
Response: According to the theoretical predictions2, which were recently supported also experimentally1, the plasmonic figure of merit due to the bulk intraband transition in Bi2Se3 leads to the plasmonic absorption in the range of 200 - 400 nm, while interband and intraband transition involving TSS yield in the plasmonic response in the range of 400 - 700 nm and above 1 μm, respectively. As described in the manuscript, both silica and Bi2Se3 show adsorption peaks in the spectral range ≈ 210 - 300 nm.
Shift in this spectral range can be related to: a) silica layer thickness or b) the effect of the silica on the plasmonic absorption resulting from the bulk intraband transition. However, since we are not sure what caused this shift, we did not want to speculate and discuss this in the manuscript.
As described in response to comment 1, the main point of the UV-vis measurements was to track the LSPR features resulting from TSS in connection to the silica layer thickness, which is the spectral range from ≈ 400 nm above. Therefore, only the ranges where absorption features appear, and not exact values of the peaks, were reported.
COMMENT 3: At page 9 the authors discuss the phototermal properties of NPs, and extensively compre photo-termal conversion behavior for the three silica-coated systems and standard however, a comparison with the EG-coated Bi2Se3 nanoparticles is neither reported or discussed. The authors should at least mention some comparison based on literature.
- Response: Complete photo-thermal investigation was done on pristine (uncoated), silica-coated and ethylene-glycol coated nanoparticles. Moreover, photo-thermal properties of ethylene-glycol coated nanoparticles are also presented in all cases on Figure 4 – green lines. As it is defined in the Experimental section under Synthesis of Bi2Se3 nanoparticles, ethylene-glycol coated nanoparticles are referred to by the name BiSe ST.
COMMENT 4: Finally, in the conclusions section (line 453) the authors affirm that "as it is shown in this work, EG coated nanoparticles are cytotoxic...". But at page 10-11, where biocompatibility studies are reported, no tests are performed on EG-coated nanoparticles. There is a reference (46) mentioned at line 433, but this is a literature study, not a finding arising by this work, as claimed in the conclusions.
Response:
- The reference in line 433 (in revised manuscript 453) was a typo, and it has been removed.
- The biocompatibility studies with ethylene-glycol coated nanoparticles were done and discussed. As it is defined in the Experimental section under Synthesis of Bi2Se3 nanoparticles, ethylene-glycol coated nanoparticles are referred to the name BiSe ST. The cytotoxicity tests are also discussed and supported with relevant references.
- Belec, B.; Ferfolja, K.; Goršak, T.; Kostevšek, N.; Gardonio, S.; Fanetti, M.; Valant, M. Rep. 2019, 9, 190571.
- Yin, J.; Krishnamoorthy, H. N. S.; Adamo, G.; Dubrovkin, A. M.; Chong, Y.; Zheludev, N. I.; Soci, C. NPG Asia Materials 2017, 9, e425.
Reviewer 4 Report
Dear Authors,
Thank you very much for a nice manuscript with good scientific content.
I have only a few comments to your manuscript, I would like you to adress.
In line 216 you write: an optical lens with a spot size of about 8 mm. This does not make sense to me. Either you mean an optical lens focusing to a spot size of 8 microns or your mean an optical lens with a focal length of 8 mm. Please clarify.
The text in the two lines 450 and 451 simply does not make sense to me, Could you re-write them?
Best regards,
The reviewer
Author Response
Reviewer 4: Dear Authors, Thank you very much for a nice manuscript with good scientific content.
I have only a few comments to your manuscript, I would like you to address.
COMMENT 1: In line 216 you write: an optical lens with a spot size of about 8 mm. This does not make sense to me. Either you mean an optical lens focusing to a spot size of 8 microns or your mean an optical lens with a focal length of 8 mm. Please clarify.
Response: This sentence was rewritten to: Laser light was focused on a quartz cuvette with a size of 1 x 1 x 3 cm using an optical lens focusing to a spot size of 8 mm.
COMMENT 2: The text in the two lines 450 and 451 simply does not make sense to me, Could you re-write them?
Response: The authors agree with the reviewer, that this sentence is not placed right and as it is, it makes no sence. Therefore, the sentence: “Nevertheless, that the EG layer has positive properties (increase colloidal and chemical stability in physiological medium), EG layer also alters/masks the optical properties resulting from TSS”, was deleted. The message of the sentence meaning was added in line 466-467, in the revised manuscript (highlighted). The added part in those lines is: …. as it was demonstrated in the case of EG coating [7].
Round 2
Reviewer 1 Report
All the questions has fully revised.